# Regulation of Flowering Time by Environmental Factors in Plants

**DOI:** 10.3390/plants12213680

**Published:** 2023-10-25

**Authors:** Zion Lee, Sohyun Kim, Su Jeong Choi, Eui Joung, Moonhyuk Kwon, Hee Jin Park, Jae Sung Shim

**Affiliations:** 1School of Biological Sciences and Technology, Chonnam National University, Gwangju 61186, Republic of Korea; zion981024@naver.com (Z.L.); s120511@naver.com (S.K.); tnwjd12266@naver.com (S.J.C.); wjddmltls296@naver.com (E.J.); 2Division of Life Science, ABC-RLRC, PMBBRC, Gyeongsang National University, Jinju 52828, Republic of Korea; mkwon@gnu.ac.kr; 3Department of Biological Sciences and Research Center of Ecomimetics, College of Natural Sciences, Chonnam National University, Gwangju 61186, Republic of Korea; 4Institute of Synthetic Biology for Carbon Neutralization, Chonnam National University, Gwangju 61186, Republic of Korea

**Keywords:** flowering time, plants, light, drought, salinity, temperature

## Abstract

The timing of floral transition is determined by both endogenous molecular pathways and external environmental conditions. Among these environmental conditions, photoperiod acts as a cue to regulate the timing of flowering in response to seasonal changes. Additionally, it has become clear that various environmental factors also control the timing of floral transition. Environmental factor acts as either a positive or negative signal to modulate the timing of flowering, thereby establishing the optimal flowering time to maximize the reproductive success of plants. This review aims to summarize the effects of environmental factors such as photoperiod, light intensity, temperature changes, vernalization, drought, and salinity on the regulation of flowering time in plants, as well as to further explain the molecular mechanisms that link environmental factors to the internal flowering time regulation pathway.

## 1. Introduction

Flowering time stands as one of the most crucial traits in plants, as it plays a pivotal role in determining reproductive success within a given habitat. To optimize reproductive outcomes, plants must trigger flowering amidst favorable environmental conditions. To accomplish this, plants have evolved sophisticated molecular sensing systems capable of recognizing the dynamic changes occurring in their habitats. Therefore, the identification and characterization of the molecular components responsible for perceiving alterations in environmental conditions are crucial to gaining insights into the precise physiological responses of plants under varying circumstances [1,2]. Flowering time in plants is primarily controlled by the accumulation of florigen [3,4]. Florigen, a systemic signal synthesized in leaves, is transported to the shoot apical meristem, thereby initiating flowering in plants [3]. In Arabidopsis plants, FLOWERING LOCUS T (FT) acts as a major florigen. Therefore, the biosynthesis of FT and its accumulation in the shoot apical meristem must be precisely regulated according to internal developmental and external environmental cues. Among the various environmental signals relevant to plants, photoperiod has garnered significance as a pivotal input that reflects the flow of seasonal changes [5,6]. The alteration in photoperiod predominantly shapes the degree of *FT* expression, mainly through the transcriptional and posttranslational regulation of CONSTANS (CO). Thus, the photoperiod-dependent regulation of CO accumulation is important to accelerate flowering under specific seasons [5]. Additionally, Arabidopsis plants harbor an alternative pathway to induce flowering, which determines the timing of floral transition independently of photoperiod. This autonomous pathway is closely related to the regulation of the *FLOWERING LOCUS C* (*FLC*) floral repressor at various levels, including transcription, RNA processing, and epigenetic controls [7,8]. In addition to photoperiod, various environmental factors jointly influence the pathways that regulate flowering time. Here, we summarize the recent discoveries mainly obtained from Arabidopsis plants explaining the molecular mechanisms that link environmental factors such as photoperiod, light intensity, temperature change, drought, and salinity to flowering time regulation pathways.

## 2. Induction of Flowering by Florigen in Plants

The molecular mechanisms governing flowering time have undergone extensive investigation in Arabidopsis plants. The timing of floral transition primarily hinges on the accumulation of the florigen FT within the shoot apical meristem (SAM). This accumulation prompts the conversion of the shoot apical meristem into a floral meristem (FM). FT belongs to the phosphatidylethanolamine binding protein (PEBP) family, alongside TWIN SISTER OF FT (TSF) and TERMINAL FLOWER 1 (TFL1) [9]. Florigen is ubiquitously present in flowering plants. For instance, rice possesses 13 *FT* homologous genes, including the well-defined florigen-encoding genes *HEADING DATE 3a* (*Hd3a*) and *RICE FLOWERING LOCUS T1* (*RFT1*) [10,11]. Maize encodes 15 FT homologs, including *CENTRORADIALIS 8* (*ZCN8*) [12]. These findings suggest that the timing of flowering is determined by the spatiotemporal accumulation of florigen in plants.

*FT* is expressed in phloem companion cells in leaves, after which it is transported to the SAM [3,13]. Although the FT protein is small enough to move passively through plasmodesmata, its movement is under the regulation of specific transporters. Particularly, FT-INTERACTING PROTEIN 1 (FTIP1), which is localized in the endoplasmic reticulum, is required for the transport of FT from phloem companion cells to sieve elements [14]. In addition to FTIP1, SODIUM POTASSIUM ROOT DEFECTIVE 1 (NaKR1)/NUCLEAR-ENRICHED PHLOEM COMPANION CELL 6 (NPCC6) is also involved in the movement of FT into the phloem stream. The loss of function of *NaKR1/NPCC6* significantly reduced FT transport to the shoot apical meristem [15]. In addition to its role as FT transporter, NaKR1/NPCC6 also controls the transcription of *FT* through the miR156-SQUAMOSA PROMOTER BINDING PROTEIN-LIKE 3 (SPL3) module [16]. Unlike *FTIP1*, whose expression is not affected by photoperiod [14], *NaKR1/NPCC6* is highly expressed under inductive long-day conditions [15]. The daylength-dependent activation of *NaKR1/NPCC6* expression is governed by CO [15]. Additionally, it has been reported that MYB transcription factor *FE*/*ALTERED PHLOEM DEVELOPMENT* (*FE*/*APL*) is required for the upregulation of both *FTIP1* and *NaKR1/NPCC6* expression as well as *FT* expression [17,18]. These results suggest that, at least at the transcriptional level, the production and transport of *FT* coordinate with each other.

Measuring the movement kinetics of FT provided concrete insights into the speed at which FT travels from companion cells to the shoot apical meristem [13]. The authors used the promoter of the heat shock-induced gene *HEAT SHOCK PROTEIN 18.2* (*HSP18.2*) to control the expression of *FT*. After transient heat treatment in a single leaf blade, the accumulation of the FT protein in the shoot apex was measured via 2D-PAGE. Through this approach, it was determined that 8 h was required for sufficient FT transport from leaves to phloem, with the FT protein becoming detectable in the SAM 12 h after heat shock treatment. These findings support the notion that FT transport is actively controlled. Additionally, it was discovered that at least three amino acid residues, V70, S75, and R83, are responsible for the active transport of FT from the leaves to the SAM [13]. Interestingly, alanine substitution of all three amino acid residues did not affect the interaction of the FT variant with FTIP1. Moreover, the aforementioned FT variant with the three amino acid mutations has been detected in phloem sap [19]. These findings suggest that the reduced transport of the FT variant to the SAM is not a result of phloem loading but rather could be attributed to unloading FT around the SAM.

Once FT reaches the shoot apical meristem, it forms a florigen activation complex with bZIP proteins and 14-3-3 proteins. The structure of this florigen activation complex was elucidated in rice plants [20]. The rice florigen activation complex is a heterohexamer composed of two rice florigens, Hd3a, two OsFD1s, and two 14-3-3 proteins. Based on their subcellular localization patterns, it has been proposed that FT transported from leaves is initially received by 14-3-3 proteins in the cytoplasm. Afterward, the complex is translocated into the nucleus to form a complex with FD [21]. Additionally, phosphorylation of OsFD1 by rice CALCINEURIN B-LIKE PROTEIN INTERACTING PROTEIN KINASE 23 (OsCIPK3) promotes the formation of the florigen activation complex with RFT1 [22]. Aside from florigen, certain PEBP proteins, such as TFL1 and RICE CENTRORADIALIS (RCN), function as anti-florigens [4]. These anti-florigens repress the activity of florigen by competing for interaction with 14-3-3 proteins [23]. Ultimately, the florigen activation complex governs the expression of multiple floral meristem identity genes, leading to the transformation of the SAM into an FM, finally culminating in flowering.

## 3. Regulation of *FT* Expression in Plants by Environmental Factors

In addition to the transport of FT from leaves to the SAM, the sufficient expression of *FT* transcripts in leaves is another important determinant of floral transition. To induce flowering under favorable environmental conditions, the expression of *FT* is tightly controlled by multiple transcriptional activators and repressors. Recent reviews provide a comprehensive understanding of the transcriptional control of *FT* across various external circumstances [5,24,25,26]. Particularly, the present review focuses on the recent achievements explaining molecular mechanisms that control *FT* expression in response to photoperiod, light intensity, temperature, drought, and salinity.

### 3.1. Photoperiod

Due to the Earth’s rotation on its tilted axis and its orbit around the sun, organisms inhabiting the planet undergo annual seasonal changes. In order to maximize their chances of survival, plants have evolved molecular mechanisms that anticipate upcoming seasonal variations by monitoring changes in photoperiod. Plants gauge photoperiodic alterations through the perception of external light conditions by multiple photoreceptors, which subsequently integrate these signals with internal circadian regulation (Song et al., 2015; Wang et al., 2021 [5,25]).

Photoperiodic flowering responses are classified into three major types: long-day, short-day, and day-neutral, based on their responses to photoperiod. The phenomenon of photoperiodic flowering has been extensively explored in Arabidopsis plants. *Arabidopsis thaliana*, classified as a facultative long-day plant, accelerates flowering under long-day conditions. This acceleration of flowering in response to long days stems from the day-length-specific regulation of *FT* expression (Figure 1). While *FT* is not substantially expressed throughout the day under non-inductive short-day conditions, its expression is markedly induced, particularly during the late afternoon, under inductive long-day conditions [27]. This day-length-dependent induction of *FT* is primarily controlled by the zinc finger-type transcription factor *CO* [28,29]. CO is a transcriptional activator that binds to the CONSTANS-responsive element (CORE) on the *FT* promoter through its C-terminal CCT domain [30]. The binding of CO to the *FT* promoter is further regulated by its physical interaction with other proteins. For example, ASYMMETRIC LEAVES 1 (AS1) and nuclear factor Y (NF-Y) interact with CO to recruit it to the *FT* promoter [31,32,33]. B-box transcription factors also regulate CO protein activity via physical interaction [34]. B-box transcription factor 28 (BBX28) forms a complex with CO and inhibits the association of CO with the *FT* promoter [35]. BBX30 and BBX31 recruit CO into a TOPLESS repressor protein. The trimeric complex represses the expression of *FT*, thus delaying the flowering process [36]. Similarly, BBX19 deactivates the CO protein through physical interaction. The expression pattern of *BBX19* is opposite to that of *CO*, suggesting that BBX19-mediated deactivation is important for the time-specific induction of *FT* expression [37].

To ensure that *FT* induction takes place during the late afternoon, CO activity must be restricted to the long afternoon. Achieving the long-day-specific accumulation of CO involves a dual mechanism: circadian clock-mediated transcriptional regulation and external light conditions. The daily oscillation of *CO* expression is predominantly orchestrated by the repressor CYCLING DOF FACTORs (CDFs) [38,39,40]. CDFs act as direct repressors of *CO* expression. Mutations in *CDF*s (*cdf1*, *cdf2*, *cdf3*, and *cdf5*) stimulate *CO* expression under both long- and short-day conditions. The repression of *CO* expression by CDFs can be attributed to their interaction with the TOPLESS transcriptional corepressor [41]. Moreover, the circadian clock governs CDF-mediated *CO* transcriptional repression [42,43,44]. In the morning, CIRCADIAN CLOCK ASSOCIATED1 (CCA1) and LATE ELONGATED HYPOCOTYL (LHY) activate *CDF* expression, whereas in the afternoon, PSEUDO-RESPONSIVE REGULATORs (PRRs) repress *CDF* expression, thus creating a diurnal rhythmic *CDF* expression pattern.

Beyond transcriptional regulation, CDF protein stability is further regulated by FLAVIN-BINDING, KELCH REPEAT, F-BOX1 (FKF1), and GIGANTEA (GI) [40]. Under inductive long-day conditions, *FKF1* and *GI* are highly expressed in the afternoon. When exposed to light, FKF1 is activated through its LOV domain in response to blue light. The activated FKF1 forms a complex with GI. This FKF1-GI complex facilitates the ubiquitin-dependent degradation of CDFs. Under short-day conditions, *FKF1* is predominantly expressed during the night, and the expression of GI is out of sync with *FKF1*, diminishing the likelihood of FKF1-GI complex formation. Therefore, the FKF1-GI complex contributes to the long-day-specific degradation of CDFs in the afternoon, ultimately inducing *CO* expression in the afternoon. Once CDFs are eliminated by the FKF1-GI complex, several transcriptional activators such as FLOWERING BHLHs (FBHs) and class II TEOSINTE BRANCHED 1/CYCLOIDEA/PROLIFERATING CELL NUCLEAR ANTIGEN FACTORs (TCPs) directly enhance *CO* expression [45,46].

In addition to the transcriptional regulation of *CO*, the stability of CO protein is intricately regulated in response to external light conditions. CO protein stability is enhanced by far-red and blue light conditions but diminished by red light and darkness [47], indicating the involvement of multiple light signaling components in CO protein stability control. Phytochrome B (PHYB) orchestrates the red light-dependent destabilization of CO [47]. This PHYB-mediated destabilization is partially explained by the physical interaction with the E3 ligase HIGH EXPRESSION OF OSMOTICALLY RESPONSIVE GENES 1 (HOS1) [48,49]. HOS1 interacts with CO, leading to its degradation in the morning. Consequently, the red light-dependent destabilization of CO by PHYB and HOS1 reduces CO accumulation in the afternoon. During the night, the CONSTITUTIVE PHOTOMORPHOGENIC 1 (COP1)-SUPPRESSOR OF PHYA-105 1 (SPA1) complex participates in CO protein degradation [50,51,52]. Moreover, the COP1-dependent degradation of CO relies on CO phosphorylation [53]. A recent study reported that SHAGGY-like kinase 12 (SK12) mediates CO phosphorylation [54]. Specifically, SK12 phosphorylates T119 of CO, leading to its destabilization. Conversely, the FK506-binding protein FKIP12 prevents the degradation of phosphorylated CO [55]. Under far-red light, Phytochrome A (PHYA) stabilizes CO [47]. An analysis conducted under natural sunlight conditions emphasized the significance of PHYA’s role in CO stabilization and *FT* expression [56]. In contrast to laboratory conditions (R/FR > 2.0), *FT* expression is considerably induced in the morning under natural conditions (R/FR = 1.0). This morning-specific *FT* expression is mediated by both CO stabilization and PHYA [56]. Further investigation is warranted to elucidate the manner in which PHYA mediates CO stabilization. Blue light-dependent CO stabilization is controlled by Cryptochrome (CRY) and FKF1. Upon activation by blue light, CRY2 forms a complex with COP1 and SPA1, thereby decreasing COP1-SPA1-mediated CO degradation [52]. Particularly, FKF1 is crucial for the afternoon accumulation of CO under inductive long-day conditions. FKF1′s diurnal expression pattern allows it to accumulate in the afternoon. FKF1 forms a complex with CO to stabilize it, and this interaction is potentiated by blue light [29]. Additionally, PRR is involved in CO stabilization. Genetic analysis has indicated that PRRs are essential for both morning and afternoon CO accumulation [57]. The stabilization of CO by FKF1 and PRR is also interconnected with COP1. For example, FKF1 hampers COP1-mediated CO degradation and COP1 homo-dimerization [58]. Similarly, CO accumulation is enhanced by the introduction of a *cop1* mutation into a *toc1 prr5 prr7 prr9* quadruple mutant [57], highlighting the role of PRRs in stabilizing CO by inhibiting COP1 activity. These intricate and interconnected light signaling pathways collectively shape the photoperiod-specific accumulation of CO during the day, thereby triggering the activation of *FT* transcription in plants.

### 3.2. Light Intensity

Light intensity also affects the timing of floral transition in plants. For example, Arabidopsis plants grown under high light intensity (800 µmolm^−2^s^−1^) flowered earlier than under normal light intensity (100 µmolm^−2^s^−1^). However, Arabidopsis accessions that contain nonfunctional alleles of *FLC* did not flower earlier under high light. Moreover, vernalization is required to accelerate flowering in Arabidopsis plants harboring a functional *FRI* allele [59]. These results suggest that *FLC* is involved in high light-induced flowering in Arabidopsis plants. As expected, high light treatment significantly reduced the expression level of *FLC*. High light-induced suppression of *FLC* is controlled by chloroplast retrograde signals. A PLANT HOMEODOMAIN-TYPE TRANSCRIPTION FACTOR WITH TRANSMEMBRANE DOMAINS (PTM) mediates chloroplast retrograde signals generated by high light intensity. PTM undergoes proteolysis under high light conditions, resulting in the accumulation of its N-terminal fragment in the nucleus [60]. The PTM N-terminal fragment physically interacts with FVE/MULTICOPY SUPPRESSOR OF IRA1 4 (MSI4). Moreover, PTM is required for high light-induced binding of FVE on *FLC* chromatin [59]. FVE/MSI4 is an Arabidopsis homolog of the retinoblastoma-associated protein that mediates suppression of *FLC* expression through histone deacetylation [61]. Low light intensity generally retards the growth of plants. In Arabidopsis, low light intensity delays vegetative phase change and floral transition [62,63]. The growth retardation is associated with an increase in miR156 and miR157, and a decrease in their *SQUAMOSA PROMOTER-BINDING PROTEIN-LIKE (SPL)* targets. miR156 and miR157 target *SPL*s for post-transcriptional degradation, which in turn delays floral transition by prolonging vegetative growth [64]. Exogenous application of sucrose partially rescued growth retardation and *MIR156*/*MIR157* expression, suggesting that a decrease in carbohydrate production under low light conditions in part causes growth retardation [62]. It has been reported that sugar content affects the timing of flowering in plants [65,66]. In Arabidopsis, exogenous sugar (sucrose, glucose, fructose, and maltose) treatments decreased the expression of *MIR156*, a repressor of floral transition. Similarly, trehalose-6-phosphate, which functions as a proxy for internal carbohydrate status in plants, also promotes flowering by repressing the expression of *MIR156* [67]. These findings suggest that chloroplastic sugar production and retrograde signals participate in the control of floral transition in response to light intensity.

### 3.3. Temperature Changes

Changes in temperature occurring during the plant lifecycle constitute another environmental cue that triggers significant alterations in flowering time in plants. Generally, high temperatures accelerate flowering, whereas low temperatures delay it [68,69]. To adapt the timing of flowering according to ambient temperature, plants have developed multiple temperature-sensing pathways that integrate temperature signals into the flowering time pathway (Figure 2).

The transcription factor PHYTOCHROME INTERACTING PROTEIN 4 (PIF4) plays a key role in the acceleration of flowering under high-temperature (27 °C) conditions [70,71]. PIF4 directly binds to the promoter of *FT* to enhance *FT* expression [71]. Elevated temperatures induce the expression of *PIF4*. The transcription of *PIF4* is activated by the BRASSINAZOLE RESISTANCE (BZR1) and TCP5 transcription factors [72,73,74]. BZR1 is known to mediate brassinosteroid signals [75]. High temperatures promote the nuclear localization of BZR1. Within the nucleus, BZR1 binds to the *PIF4* promoter, thus activating its expression [72]. TCP5 positively regulates both the transcription and activity of PIF4 under high temperatures [73]. Other class II TCPs also participate in the regulation of flowering time. For instance, TCP13 and TCP17 directly activate the expression of the floral meristem identity gene APETALA1 (AP1) [76]. TCP3 and TCP4 act as positive regulators of *CO* transcription [45]. Recent reports have also demonstrated the involvement of PIF4 and TCP4 in the high temperature-mediated restriction of cell division [77]. PIF4 forms a complex with TCP4 to regulate the expression of the cell cycle inhibitor *KIP-RELATED PROTEIN1* (*KRP1*). Similarly, TCP13 negatively regulates leaf cell expansion by suppressing the expression of *ARABIDOPSIS THALIANA HOMEOBOX 12* (*ATHB12*) [78]. Exploring the roles of other TCP members in high-temperature flowering could yield intriguing insights.

The evening complex (EC), consisting of EARLY FLOWERING 3 (ELF3), ELF4, and LUX ARRHYTHMO (LUX), not only regulates temperature-dependent transcription but also impacts the activity of PIF4 [79,80,81]. Operating as an oscillator of the circadian clock, the EC generates diurnal rhythmic PIF4 expression by repressing *PIF4* expression [79]. Higher temperatures diminish the DNA binding activity of LUX4, leading to the de-repression of EC-dependent *PIF4* repression [80,81]. In addition to transcriptional control, ELF3 attenuates the DNA binding activity of PIF4 through physical interaction [82]. ELF3 forms a speckle within the nucleus at high temperatures, whereas it is diffused under low temperatures. This temperature-dependent phase transition, possibly mediated by a prion-like domain, suggests that ELF3 acts as a temperature sensor [83]. These three mechanisms, governed by the EC, work in coordination to regulate the temperature-dependent role of PIF4.

In addition to the alteration of transcriptional regulation, histone modification plays a crucial role in the temperature-dependent regulation of flowering. H2A.Z, a histone H2 variant, exhibits a stronger DNA binding affinity than H2 and can impede the access of transcription factors to their target sites [84]. In Arabidopsis, three genes encode H2A.Z: *HISTONE H2A PROTEIN 8* (*HTA8*), *HTA9*, and *HTA11* [85]. Mutations in both *HTA9* and *HTA11* lead to early flowering, accompanied by upregulation of *FT* expression [86], underscoring the role of H2A.Z integration as a negative regulator of floral transition. H2A.Z incorporation into nucleosomes undergoes alterations based on ambient temperatures. For instance, H2A.Z occupancy near the transcription start site of FT is diminished under high temperatures [87], increasing the accessibility of transcriptional activators to the *FT* promoter. Beyond H2A.Z incorporation, histone deacetylation is equally pivotal for temperature-dependent flowering. POWERDRESS (PWR), a SANT-domain-containing protein, interacts with HISTONE DEACETYLASE 9 (HDA9). A mutation in *PWR* attenuates high temperature-induced hypocotyl elongation and flowering [88]. PWR is required for deacetylation of H3K9 at the +1 nucleosome of *PIF4* and its target *YUCCA8* (YUC8). Furthermore, increased deacetylation of *FT* chromatin is observed under inductive long-day conditions [89]. A deeper exploration of how temperature influences *FT* chromatin status would provide more comprehensive insights into the molecular regulatory mechanisms of these phenomena.

Apart from PIF, MADS-box-containing transcription factors in Arabidopsis, namely SHORT VEGETATIVE PHASE (SVP) and FLOWERING LOCUS M (FLM), assume pivotal roles in temperature-dependent flowering. SVP functions as a negative regulator of *FT* expression [90]. Moreover, temperature influences both the stability and activity of SVP. Higher temperatures induce 26S proteasome-mediated degradation of SVP, thereby releasing the repression on *FT* transcription [91]. Temperature-dependent regulation of SVP activity is further modulated by the alternative splicing of *FLM* [92,93]. In Arabidopsis, alternative splicing of *FLM* yields two isoforms: *FLM-β* and *FLM-δ* [92]. These variants exert opposing functions in regulating flowering time. Specifically, *FLM-β* acts as a negative regulator, whereas *FLM-δ* functions as a positive regulator of flowering [92]. Low temperatures elevate the *FLM-β/FLM-δ* ratio, while high temperatures diminish it [91,94]. The temperature-dependent alternative splicing of FLM is mediated by RNA-binding proteins GLYCINE-RICH RNA-BINDING PROTEIN 7 (GRP7) and GRP8 [95]. Additionally, CYCLIN-DEPENDENT KINASE G2 (CDKG2), in conjunction with CYCLIN L1 (CYCL1), modulates the alternative splicing of FLM. In the *cdkg2 cycl1* double mutant, *FLM-β* transcript levels decrease, while *FLM-δ* transcript levels rise significantly across the range of ambient temperatures [96]. Histone H2 lysine 36 trimethylation (H3K36me3) is also implicated in temperature-dependent alternative splicing and flowering time regulation [97]. Pajoro et al. (2017) identified a link between H3K36me3 and ambient temperature-dependent flowering. Genes that undergo temperature-dependent alternative splicing exhibit an enrichment of H3K36me3. The absence of the *SDG8* and *SDG26* methyltransferases causes changes in the alternative splicing of FLM under high temperatures.

### 3.4. Vernalization

Vernalization is the process by which the flowering of plants is promoted by prolonged exposure to the cold of winter [98]. In plants, vernalization suppresses the expression of a gene that encodes the repressor of flowering. Differently than with cold acclimation, vernalization is not triggered by short-term cold exposure. Rather, long-term cold exposure triggers epigenetic changes during the winter season to establish stable changes that remain until the following spring, resulting in the acceleration of flowering the following year [98,99]. Arabidopsis plants can be divided into summer-annual or winter-annual plants based on their requirement of vernalization [100]. Genetic analysis found *FLC* and *FRIGIDA* (*FRI*) as major components involved in flowering time regulation by vernalization. Both FLC and FRI are repressors of flowering. FRI negatively regulates flowering time by upregulating the expression of *FLC* [101]. Winter-annual Arabidopsis plants have functional *FRI*, whereas summer-annual Arabidopsis plants have a genomic deletion of *FRI* [102], leading to early flowering in summer-annual plants. Prior to vernalization, *FLC* is highly expressed in plants to prevent flowering. A prolonged exposure to the cold of winter represses the expression of *FLC,* which in turn releases the repression of *FT* and *SUPPRESSOR OF OVEREXPRESSION OF CO 1* (*SOC1*) [103,104]. Therefore, stable suppression of *FLC* by vernalization is important for floral transition in the following spring. Vernalization induces the silencing of *FLC* expression by epigenetic regulation. Epigenetic suppression of FLC by vernalization is mediated by epigenetic changes governed by Polycomb group proteins (PcG). PcGs are multi-protein complexes that control the epigenetic status of genes [105]. During vernalization, the POLYCOMB REPRESSIVE COMPLEX 2 (PRC2) complex is enriched at *FLC* chromatin to induce histone H3 lysine 27 trimethylation (H3K27me3) [104,106]. Two PLANT HOMEODOMAIN (PHD) proteins VERNALIZATION INSENSITIVE 3 (VIN3) and VIN3-LIKE 1 (VIL1)/VERNALIZATION 5 (VRN5) [107] join the core PRC2 complex (PHD-PRC2) to increase histone methylation of *FLC* during the cold. PHD-PRC2-mediated H3K27me3 is limited to the junction of the first exon with the first intron of *FLC* [108,109]. This selective histone methylation can be explained by the existence of a *cis*-regulatory DNA element on *FLC* and two *trans*-acting epigenetic readers. The *cis*-regulatory DNA element, also called cold memory element, is recognized by two epigenetic readers: VP1/ABI3-LIKE 1 (VAL1) and VAL2 [110,111]. VAL1 and VAL2 directly interact with the PRC2 complex to recruit PHD-PRC2 to *FLC*, leading to an accumulation of H3K27me3 at this region. In addition to the PHD2-PRC2 complex, noncoding RNAs produced from the *FLC* locus participate in *FLC* silencing. COLD ASSISTED INTRONIC NONCODING RNA (COLDAIR) and COLD OF WINTER-INDUCED NONCODING RNA FROM THE PROMOTER (COLDWRAP) transcribed from between the 5′ proximal promoter and the first exon, mediate *FLC* silencing by forming a repressive intragenic chromatin loop at the *FLC* locus during vernalization [112,113].

### 3.5. Drought

In response to drought conditions, plants have evolved a range of physiological, morphological, and biochemical adaptation mechanisms, which can be broadly categorized into drought avoidance, drought tolerance, and drought escape strategies [114,115]. Drought avoidance involves regulating water loss through stomatal closure and accumulating water-preserving metabolites to enhance water storage capacity. Drought tolerance mechanisms aim to maintain physiological activity under drought conditions. Drought escape is a strategy wherein plants accelerate their developmental processes to complete their life cycle before the onset of drought. If drought occurs at the early stage of vegetative growth, it can cause a strong negative effect on plant growth. Under these conditions, plants are unable to survive unless they successfully induce a drought tolerance mechanism. Once plants successfully acclimate to drought, they accelerate drought escape responses to minimize their exposure to the stress conditions [116]. Therefore, the consequences of drought in plants can be different based on the severity and duration of drought. Moreover, the impacts of drought on flowering time are different depending on the plant species, growing season, and developmental stages [117]. For these reasons, drought can act both as a positive and negative signal to induce flowering in plants. However, many plant species can promote flowering or post-anthesis development in conditions of terminal drought, indicating that drought escape is a universal characteristic of plant acclimation. In Arabidopsis, mild drought triggers the acceleration of flowering under inductive long-day conditions through the activation of *FT* expression. However, under non-inductive short-day conditions, drought leads to a delay in floral transition coupled with an increase in *FLC* expression [118]. Genetic analysis has revealed that drought-induced early flowering under inductive long-day conditions is mediated by GI [118]. Interestingly, unlike the *gi* and *ft tsf* double mutant plants, *co* mutant plants exhibit early flowering under drought conditions. Different from the flowering phenotype of the *co* mutant, *CO* is also required for the activation of *FT* expression under drought conditions [119]. This discrepancy can be explained by the role of *TSF*. Similar to *FT*, *TSF* expression is also induced by drought treatments [119]. However, the drought-induced expression of *TSF* is disrupted in the *gi* mutant but not in the *co* mutant. This suggests that drought triggers early flowering through the upregulation of *FT* via the GI-CO pathway and the activation of *TSF* through a GI-dependent pathway (Figure 3).

Drought induces various physiological responses in plants through both ABA-dependent and ABA-independent pathways [120]. Notably, the drought-induced early flowering appears to be connected to the ABA signaling pathway. While the role of ABA in flowering time regulation has been contentious due to its varying effects across plant species [121], it acts as a positive regulator at least in the context of drought-induced early flowering. Exogenous ABA treatment accelerates flowering [122], and ABA biosynthesis mutants (*aba1* and *aba2*) exhibit delayed flowering under both normal and drought conditions. Furthermore, *aba1* mutant plants display reduced sensitivity to drought treatment in terms of flowering time compared to wild-type plants [118]. This is supported by gene expression analyses, which demonstrate that drought fails to induce the expression of *FT* and *TSF* in a*ba1* mutant plants. Additionally, ABI5-BINDING PROTEINs (AFPs) modulate the transcriptional activity of CO. By attenuating ABI5 activity, AFPs negatively regulate ABA signaling [123]. Interestingly, AFPs also act as negative regulators of flowering. Among them, AFP2 physically interacts with CO to diminish its transcriptional activity. This interaction involves recruiting the transcriptional corepressor TPR2 through the EAR domain of AFP2 [124]. These findings collectively suggest that the drought-mediated accumulation of ABA accelerates flowering, particularly under inductive long-day conditions.

The transduction of drought signals to flowering involves several transcription factors. For instance, ABA-responsive element (ABRE)-binding factors (ABFs) play a pivotal role in regulating flowering under drought conditions. The *abf3 abf4* double mutant fails to display ABA-mediated early flowering. Moreover, the *abf2 abf3 abf4* mutant exhibits delayed flowering along with reduced expression of *CO* and *FLOWERING BHLH 3 (FBH3)* [125]. Further exploration through gene expression analysis has revealed that ABF3 and ABF4 induce flowering by activating the expression of *SUPPRESSOR OF OVEREXPRESSION OF CONSTANS* (*SOC1*), rather than *FT*. Interestingly, ABF3 and ABF4 do not directly bind to the *SOC1* promoter. Instead, their physical interaction with NF-YC subunits is essential for the upregulation of *SOC1* expression [122]. Additionally, ABA-dependent phosphorylation enhances the activities of ABFs. RXXS/T sites within ABFs are phosphorylated by the ABA-dependent SNF1-related kinase 2 (SnRK2), leading to ABF stabilization [125,126]. This phosphorylation aids in the increased stability of ABFs [127]. Similar to Arabidopsis, mild drought also accelerates flowering in rice by activating the expression of its two florigens (*Hd3a* and *RFT1*) [128]. This drought-induced promotion of flowering can be attributed to bZIP transcription factors [128,129,130]. For example, OsbZIP23 upregulates *Early heading date 1*(*Ehd1*) expression and downregulates *Grain number, plant height, and heading date 7* (*Ghd7*) expression to facilitate flowering under drought conditions [128]. Ehd1 promotes flowering by upregulating the expression of *Hd3a* and *RFT1*. By contrast, Ghd7 negatively regulates *Ehd1* expression [131,132]. Furthermore, *OsFD1/OsbZIP77*, the product of which forms a flowering activation complex with Hd3a and RFT1, is upregulated by ABA treatment [129]. In addition to bZIP transcription factors, previous studies have reported that the NAM, ATAF1/2, and CUC2 (NAC) domain transcription factors VASCULAR PLANT ONE-ZINC FINGERs (VOZs) also play an important role in drought-mediated regulation of flowering time in tomato. The function of VOZs in flowering time regulation has been extensively studied in Arabidopsis [133,134]. VOZ1 and VOZ2 were originally identified as phytochrome-interacting proteins through yeast two-hybrid screening [133,134]. Notably, the *voz1 voz2* double mutant in Arabidopsis displays delayed flowering under long-day conditions. The delayed flowering phenotype of *phyb* mutants is suppressed by the *voz1 voz2* mutation, indicating that VOZ1 and VOZ2 specifically participate in PHYB-mediated flowering time regulation [133]. The mutation of *voz1* and *voz2* reduces *FT* expression while increasing *FLC* expression [134]. Interestingly, genetic analyses have shown that the late flowering of the *voz1 voz2* mutant is unaffected by the *flc* mutation, suggesting that VOZs primarily regulate *FT* expression and flowering time independently of *FLC* [134]. Despite the absence of a VOZ binding element in the *FT* promoter, genetic studies reveal that CO is necessary for VOZs-mediated regulation of *FT* expression, and biochemical analysis shows that VOZ1 and VOZ2 physically interact with CO [134]. This interaction suggests that VOZ-CO binding stabilizes CO, leading to the upregulation of *FT*. This mechanism aligns with other instances of phytochrome-interacting proteins in plants. For example, PHYTOCHROME DEPENDENT LATE FLOWERING (PHL) stabilizes CO by counteracting PHYB’s inhibitory effect [135]. PHL forms a complex with both PHYB and CO, which mitigates PHYB-mediated CO destabilization. Furthermore, the involvement of VOZs in drought-induced flowering is suggested through ABA-dependent phosphorylation regulation [136]. In tomatoes, the signaling pathway involving OPEN STOMATA 1 (SlOST1) –SlVOZ1 is crucial for the regulation of flowering time under drought conditions [136]. Phosphoproteomic analyses identified SlVOZ1 as a phosphorylation substrate of SlOST1, and further in vitro analysis confirmed that SlOST1 interacts with SlVOZ1 and phosphorylates it. VOZ proteins are primarily localized in the cytoplasm but function within the nucleus. Phosphorylation of SlVOZ1 enhances its stability and nuclear accumulation. Subsequent DNA affinity purification sequencing and ChIP analysis demonstrated that SlVOZ1 is physically associated with the promoter of the tomato *FT* ortholog *SINGLE FLOWER TRUSS (SFT)* [136]. These results suggest that the SlOST1–SlVOZ1 interaction is involved in ABA-mediated drought escape responses (early flowering under drought conditions) (Figure 3). Interestingly, OST1/SnRK2.6 also phosphorylates FBH3, reducing its DNA binding activity by promoting monomer formation [137]. Furthermore, OST1/SnRK2.6 phosphorylates PHYB, negatively regulating red light responses [138]. These findings indicate that OST1/SnRK2.6 is a major component in modulating flowering time under drought conditions. In addition to phosphorylation, the stability of VOZ1 is controlled by the 26S proteasome. Previous reports indicate that the BRUTUS (BTS) E3 ligase degrades VOZ1 and VOZ2 in the nucleus [139]. BTS accumulates under drought and low temperatures, with its level being negatively correlated with that of VOZ2. Therefore, exploring the function of BTS in drought-mediated regulation of flowering time in plants represents an intriguing avenue for research.

Splicing of *FLC* is also implicated in drought-induced flowering [140,141,142]. In Arabidopsis, a mutation in the splicing factor *AtU2AF65b* led to early flowering under both long- and short-day conditions. The acceleration of flowering in the mutant can be attributed to reduced expression of *FLC* due to increased intron retention and decreased transcription. Besides *FLC*, AtU2AF65b also affects the expression of four *FLC* paralogs (*MADS AFFECTING FLOWERING 1* (*MAF1*), *MAF2*, *MAF3*, *MAF4*) [140]. ABA did not accelerate flowering in the *atu2af65b* mutant plants [140], suggesting the potential involvement of *AtU2AF65b* in ABA-mediated flowering time regulation (Figure 3).

*PHYTOCHROME AND FLOWERING TIME 1*/*MEDIATOR 25* (*PFT1/MED25*) also regulates flowering time under drought conditions [143]. PFT1/MED25 is a subunit of the mediator complex [144], which is required for transcription by RNA polymerase II, and a subunit of the complex relays information from cellular signals and transcription factors to the RNA polymerase II [145]. As its name implies, PFT1/MED25 was initially identified as a regulator of flowering time downstream of PHYB [146]. PFT1/MED25 promotes flowering through both CO-dependent and independent pathways [147]. PFT1/MED25 activates *CO* and *FT* transcription, in addition to activating *FT* expression independently of CO. Interestingly, the activity of PFT1/MED25 is controlled by activation by destruction [148]. Activation by destruction is a counterintuitive explanation for the phenomenon where the degradation of transcriptional activators increases their functions [149]. The turnover of PFT1/MED25 by MED25-BINDING RING-H2 PROTEIN1 (MBR1) and MBR2 induces the expression of *FT* [148]. A yeast two-hybrid screening revealed that PFT1/MED25 physically interacts with DROUGHT RESPONSIVE ELEMENT BINDING PROTEIN 2A (DREB2A), a regulator of stress response and flowering [143]. Therefore, drought acts as a potent environmental signal that integrates across multiple layers of the central flowering time pathway.

### 3.6. Salinity

High salinity significantly affects plant growth and development [150,151]. Regarding flowering time, salt stress serves as a negative factor. Inhibition of growth and development of plants by high salinity may indirectly cause a delay in floral transition in plants. On the other hand, the expression of flowering genes is also controlled by salinity in plants. Here, we discuss the impacts of high salinity on the flowering time pathway in Arabidopsis. High salinity delays flowering time in Arabidopsis in a dose-dependent manner by suppressing the expression of *CO* and *FT* [152]. The delayed transition to flowering due to high salinity can be attributed to the regulation of the photoperiodic flowering pathway comprising the GI-CO-FT module [152,153,154]. Loss of function mutation of GI or CO did not induce significant differences in the flowering time between normal and high salinity conditions [152,153]. Furthermore, the flowering delay under high salinity conditions was mitigated by overexpressing GI in Arabidopsis [153]. These results suggest that GI is an important molecular player integrating high salinity into the photoperiodic flowering pathway. The decrease in *CO* and *FT* expression is explained by the destabilization of GI under saline conditions [153]. MG132 treatment reduced the degradation of GI under high salinity conditions, indicating that salt-induced GI degradation is mediated by the 26S proteasome. Additionally, GI negatively regulates salt tolerance in plants by modulating the salt overly sensitive (SOS) pathway [153,154]. The SOS pathway is a master regulatory system that maintains ion homeostasis under high salinity conditions, comprising SOS1 Na^+^/H^+^ antiporter, SOS2 kinase, and Ca^2+^-activated SOS3 [155]. Under normal conditions, GI interacts with SOS2 to inhibit SOS2-mediated phosphorylation of SOS1 [153]. Unphosphorylation of SOS1 reduces its stability and transport activity, thus diminishing the regulation of ion homeostasis in saline conditions [153,156]. High salinity triggers the destabilization of GI, leading to the liberation of SOS2. This freed SOS2 then interacts with SOS3 to activate SOS1 [155]. Moreover, the GI-SOS pathway is also involved in the regulation of flowering time. While high salinity prompts the degradation of cytoplasmic GI, nuclear GI remains stable under saline conditions [154]. The stabilization of nuclear GI is achieved through its physical interaction with SOS3. Within the nucleus, SOS3 interacts with and stabilizes GI and FKF1, thus promoting the expression of *CO* and *FT*. These regulatory mechanisms establish molecular connections between high salinity acclimation and the photoperiodic flowering pathway in plants (Figure 4).

Arabidopsis CYCLIN-DEPENDENT KINASE G2 (CDKG2) acts as a negative regulator of salinity responses in plants. A mutation in *CDKG2* led to increased salt tolerance and upregulation of several salt stress-responsive genes including *SOS1*, *SOS2*, *SOS3*, *ABI2*, and *ABI3*. Furthermore, the expression of *FT* is upregulated in c*dkg2* mutant plants, leading to early flowering [157]. Further investigation is necessary to elucidate the involvement of mRNA splicing in the SOS pathway and the regulation of flowering time.

## 4. Perspectives

Over the past 30 years, a combination of genetic and biochemical approaches has provided insights into how environmental factors influence flowering time regulation. Through these studies, it has become evident that several environmental cues are integrated into the central flowering pathway. Despite these advancements, numerous pivotal questions remain unresolved regarding the regulation of flowering time by environmental factors. For instance, drought, depending on its severity and duration, can act both as a positive and negative signal to induce flowering in plants. Similar paradoxical effects on flowering time have also been observed with ABA [26]. To decipher these perplexing phenomena, it becomes crucial to understand the origins of these contradictory effects of drought and ABA, as well as the mechanisms that underlie these effects. Does the plant possess the ability to discern the source of the increase in ABA, or is there an additional sensing system that gauges the severity of drought? Precise elucidation of the alterations in flowering time due to drought necessitates further investigation. Unlike drought, salinity usually serves as a negative factor influencing floral transition. However, the stabilization of GI in the nucleus under salinity conditions suggests that plants possess intricate and interconnected pathways to sustain their capacity to induce flowering under severe salinity conditions. Additional studies are thus needed to determine whether there might be additional components that uphold the flowering time pathway under salinity conditions. Ambient temperature is a potent environmental cue that determines the timing of flowering. While previous studies predominantly concentrated on identifying molecular components regulating *FT* expression in response to ambient temperature, recent discoveries have indicated that the movement of FT protein is also regulated by its lipid binding ability. The interaction of FT with phosphatidylglycerol results in the sequestration of FT within phloem companion cells [158]. Particularly, this sequestration of FT holds significance for temperature-dependent flowering. The mutation of *PHOSPHATIDYLGLYCEROLPHOSPHATATE SYNTHASE 1* notably increases the soluble form of FT only under low temperatures (16 °C) [158]. These reports suggest that the FT movement is also governed by ambient temperature. Furthermore, the phase transition of regulatory components might represent an additional mechanism for regulating temperature-dependent flowering time [83]. Further research into alterations in florigen movement or the contributions of known regulators in response to ambient temperature would provide key insights into the regulatory mechanisms governing temperature-dependent flowering.

## Figures and Tables

**Figure 1 plants-12-03680-f001:**
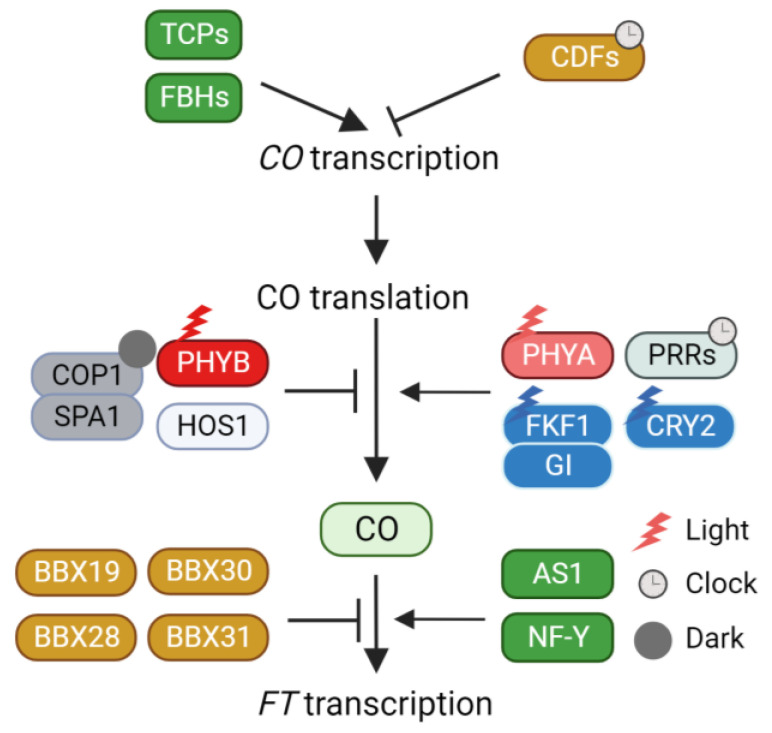
Regulation of CO activity through CO transcription, protein stability, protein interaction. CO, CONSTAMS; TCPs, TEOSINTE BRANCHED 1/CYCLOIDEA/PROLIFERATING CELL NUCLEAR ANTIGEN FACTORs; FBHs, FLOWERING BHLHs; CDFs, CYCLING DOF FACTORs; PHYB, Phytochrome B; HOS1, HIGH EXPRESSION OF OSMOTICALLY RESPONSIVE GENES 1; COP1, CONSTITUTIVE PHOTOMORPHOGENIC 1; SPA1, SUPPRESSOR OF PHYA-105 1; PHYA, Phytochrome A; PRRs, PSEUDO-RESPONSIVE REGULATORs; FKF1, FLAVIN-BINDING, KELCH REPEAT, F-BOX1; GI, GIGANTEA; CRY2, Cryptochrome 2; BBX, B-BOX TRANSCRIPTION FACTOR; AS1, ASYMMETRIC LEAVES 1; NF-Y, NUCLEAR FACTOR-Y; FT, FLOWERING LOCUS T.

**Figure 2 plants-12-03680-f002:**
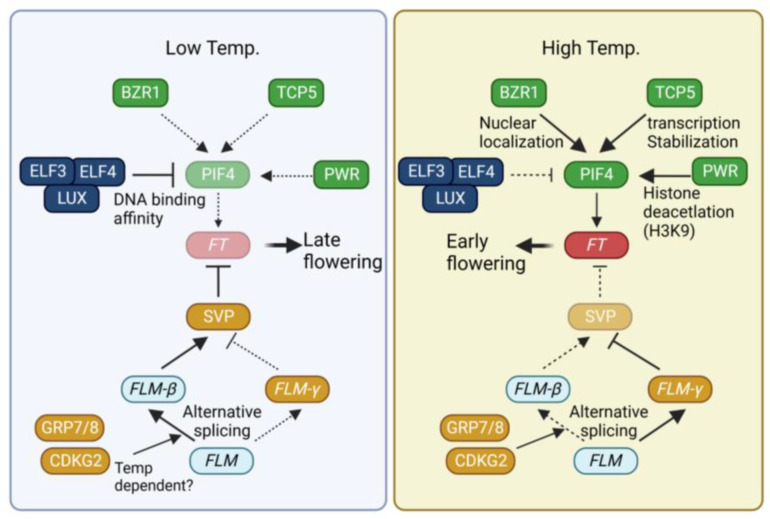
Regulation of flowering time by temperature. Description near arrow indicates detailed regulatory mechanism. Dotted line, weak contribution; solid line, strong contribution. BZR1, BRASSINAZOLE RESISTANCE 1, TCP5, TEOSINTE BRANCHED 1/CYCLOIDEA/PCF 5; PIF4, PHYTOCHROME INTERACTING PROTEIN 4; ELF3, EARLY FLOWERING 3; LUX, LUX ARRHYTHMO; PWR, POWERDRESS; FT, FLOWERING LOCUS T; SVP, SHORT VEGETATIVE PHASE; FLM, FLOWERING LOCUS M; GRP7, GLYCINE-RICH RNA-BINDING PROTEIN 7; CDKG2, CYCLIN-DEPENDENT KINASES G2.

**Figure 3 plants-12-03680-f003:**
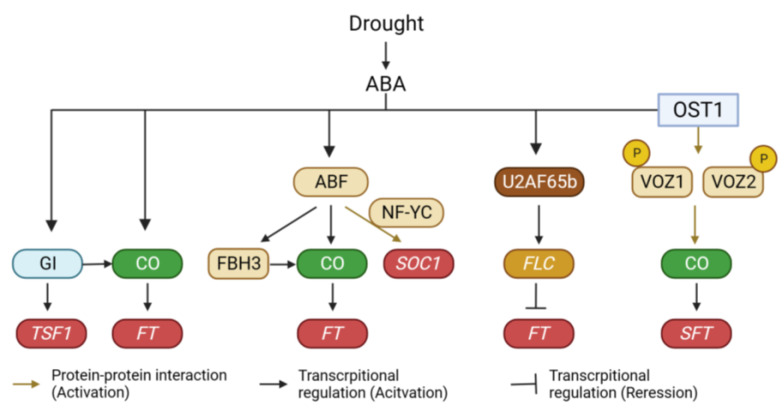
Regulation of flowering time under drought conditions. Drought acts as a positive signal to induce flowering in plants. Drought signal is incorporated into multiple layers of the flowering time regulatory pathway. GI, GIGANTEA; CO, CONSTANS; TSF1, TWIN SISTER OF FT 1; FT, FLOWERING LOCUS T; ABF, ABA-RESPONSIVE ELEMENT (ABRE)-BINDING FACTOR; NF-YC, NUCLEAR FACTOR Y SUBUNIT C; FBH3, FLOWERING BHLH 3; SOC1, SUPPRESSOR OF OVEREXPRESSION OF CO 1; FLC, FLOWERING LOCUS C; OST1, OPEN STOMATA 1; VOZ1, VASCULAR PLANT ONE-ZINC FINGER 1.

**Figure 4 plants-12-03680-f004:**
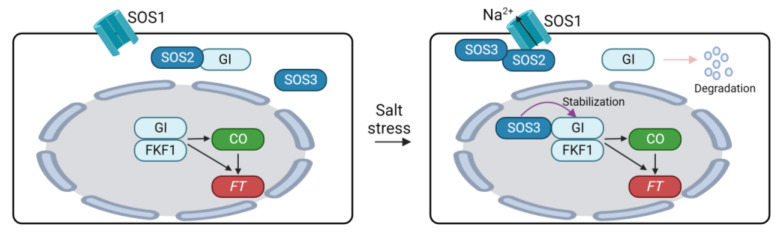
Regulation of flowering time through GIGANTEA under salinity conditions. SOS1, SALT OVERLAY SENSITIVE 1; GI, GIGANTEA; FKF1, FLAVIN-BINDING, KELCH REPEAT, F-BOX1; CO, CONSTANS, FT, FLOWERING LOCUS T.

## Data Availability

The data presented in this study are available in this article.

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
