# Peer review of "Regulation of Flowering Time by Environmental Factors in Plants"

_plants, 2023, doi:10.3390/plants12213680_

Round 1
Reviewer 1 Report
The contents of this manuscript are not enough as a review article of the effects of stresses on the regulation of flowering time. Firstly, four first pages (1, 2 and 3-1 sections) are written about photoperiodic flowering almost, and do not mention about the stresses. Secondly, only drought, salt and temperature are treated as stress although there are many other stress factors. Thirdly, the persuasive power to let readers believe that the phenomena that are introduced here are really ones influenced by stress is weak. That is because clear definition of the stress is not written.
3-2. Drought stress: This part is OK.
Fig.1: Is the transcription regulation of the downstream of U2AF65b Repression? May be Activation.
3-3. Salt stress: Salt stress serves as a negative factor in general. Inhibition of growth and development by salt stress may indirectly delay flowering time. Can such an indirect late flowering be put in a category of stress-induced late flowering? Can the late flowering be considered as stress adaptation (line 371).
3-4. Temperature: Every biochemical reaction in normal plant life is regulated by temperature. Therefore, flowering time is advanced or delayed under high or low temperature condition. It is unnecessary to consider that such a condition is stress condition. The cited reference #99 does not say that the high temperature is stress, and it does not use the term of high temperature-induced flowering. If the authors intend to consider temperature to be a stress factor, it is necessary to show what stress is. Without a clear definition of stress, it is difficult to judge if the phenomena written here are influence of the stress.
Author Response
Reviewer’s comment 1: The contents of this manuscript are not enough as a review article of the effects of stresses on the regulation of flowering time. Firstly, four first pages (1, 2 and 3-1 sections) are written about photoperiodic flowering almost, and do not mention about the stresses. Secondly, only drought, salt and temperature are treated as stress although there are many other stress factors. Thirdly, the persuasive power to let readers believe that the phenomena that are introduced here are really ones influenced by stress is weak. That is because clear definition of the stress is not written.
Response to reviewer’s comment: We appreciate your critical comment. We agree that contents in the original manuscript is not well organized. Moreover, photoperiod and temperature (In this case, temperature means fluctuation of temperature during plant’s life cycle) are not stress factors et all. It is also correct that stress-induced flowering is a response that accelerate flowering under stress conditions to complete their life cycle. We’ve also additionally described impact of light intensity and vernalization on flowering time based on other reviewer’s suggestions, and reorganized contents. Based on these problems, we think, to cover our content properly, we need to change title of the manuscript. We believe “Regulation of flowering time by environmental factors in plants” would be better title. We hope you agree with the title we’ve changed.
Reviewer’s comment 2: Fig.1: Is the transcription regulation of the downstream of U2AF65b Repression? May be Activation.
Response to reviewer’s comment: We fixed wrong information based on reviewer’s suggestion.
Reviewer’s comment 3: 3-3. Salt stress: Salt stress serves as a negative factor in general. Inhibition of growth and development by salt stress may indirectly delay flowering time. Can such an indirect late flowering be put in a category of stress-induced late flowering? Can the late flowering be considered as stress adaptation (line 371).
Response to reviewer’s comment: We agree with reviewer’s point. To clarify that salt stress can delay floral transition through inhibition of growth and development, we’ve added following description (line 538-543)
High salinity significantly affects plant growth and development [149,150]. Regarding flowering time, salt stress serves as a negative factor. Inhibition of growth and development of plants by high salinity may indirectly cause delay of floral transition in plants. On the other hands, expression of flowering genes is also controlled by salinity in plants. Here, we discuss impacts of salt stress on flowering time pathway in Arabidopsis.
Reviewer’s comment 4: 3-4. Temperature: Every biochemical reaction in normal plant life is regulated by temperature. Therefore, flowering time is advanced or delayed under high or low temperature condition. It is unnecessary to consider that such a condition is stress condition. The cited reference #99 does not say that the high temperature is stress, and it does not use the term of high temperature-induced flowering. If the authors intend to consider temperature to be a stress factor, it is necessary to show what stress is. Without a clear definition of stress, it is difficult to judge if the phenomena written here are influence of the stress.
Response to reviewer’s comment: As we mentioned above, we hope to change title to “Regulation of flowering time by environmental factors in plants”. In addition, we clearly stated that “temperature” in this session is “change in temperature occurred during plant life cycle”. We’ve provide that high temperature routinely used in Arabidopsis research is 27⁰C. We’ve also remove high temperature-induced flowering.
Please check revised manuscript from the attachment.

Reviewer 2 Report
This comprehensive review provides valuable insights for a broad audience of plant biologists, including those with interests in flowering and stress biology. It is well-crafted and thoughtfully structured, with no significant issues apparent.
Author Response
This comprehensive review provides valuable insights for a broad audience of plant biologists, including those with interests in flowering and stress biology. It is well-crafted and thoughtfully structured, with no significant issues apparent.
We appreciate your efforts for our manuscript. Based on other reviewer’s suggestions, we’ve added descriptions about the control of flowering time by light intensity and vernalization. Please check the revised manuscript form the attachment.

Reviewer 3 Report
The review entitled Regulation of flowering time in response to environmental stresses in plants was written in scientific language and I recommend publication in the present form
the engish language with minor error
Author Response
We appreciate your efforts for our manuscript. Based on other reviewer’s suggestions, we’ve added descriptions about the control of flowering time by light intensity and vernalization. The revised manuscript can be found from the attachment.

Reviewer 4 Report
This review summarized the effects of environmental stresses such as drought, salt, and ambient temperature on the flowering regulation, as well as the underlying mechanisms. These information may be helpful to the related researchers, I only have two concerns:
One is the Photoperiod pathway should address the different flowering responses into three major types: long day (LD), short day (SD), and day neutral (DN);
The other is the Temperature pathway should also include the Vernalization part.
see above.
Author Response
This review summarized the effects of environmental stresses such as drought, salt, and ambient temperature on the flowering regulation, as well as the underlying mechanisms. These information may be helpful to the related researchers, I only have two concerns:
One is the Photoperiod pathway should address the different flowering responses into three major types: long day (LD), short day (SD), and day neutral (DN);
The other is the Temperature pathway should also include the Vernalization part.
Response to reviewer’s comment: Based on you and other reviewer’s suggestions, we’ve added descriptions about the control of flowering time by light intensity and vernalization. In addition, we’ve added sentence to address the different photoperiodic flowering responses in plants as below.
Photoperiodic flowering responses are classified into three major types: long-day, short-day, and day-neutral, based on their responses to photoperiod. The phenomenon of photoperiodic flowering has been extensively explored in Arabidopsis plants. Arabidopsis, classified as a facultative long-day plant, accelerates flowering under long-day conditions.
The revised version can be found from the attachment.

Round 2
Author Response
Dear the reviewer,
We've revised our manuscript based on your comments as below. We appreciate all your efforts to improve quality of our manuscript. You can also find our revised manuscript from the attachment.
Reviewer’s comment 1. The term of salt stress is still used in the section 3-6. It should be replaced with ‘high salinity’.
Response to reviewer’s comment: We’ve replaced ‘salt stress’ with ‘high salinity’ in the section 3-6.
Reviewer’s comment 2. The term of Arabidopsis is used both in Italic form and not Italic form throughout the manuscript. If it was used as English name, it should be written in not Italic form. If it was used as scientific name, it should be written in Italic form together with species name, thaliana.
Response to reviewer’s comment: We’ve changed ‘Arabidopsis’ in non-italic form.
Reviewer’s comment 3. References cited are not shown in numerals (lines 149-150).
Response to reviewer’s comment: We’ve added reference information to the sentence.
Reviewer’s comment 4. There are some mistakes in hyphenation (line 43), abbreviation (line 73), space between words (lines 254, 255, 283), italicization of names of genes/proteins (lines 78, 483), and spelling (line 378),
Response to reviewer’s comment: We’ve carefully check our manuscript again and revised mistakes including the typos suggested by the reviewer.
